# Mechanical and Functional Improvement of β-TCP Scaffolds for Use in Bone Tissue Engineering

**DOI:** 10.3390/jfb14080427

**Published:** 2023-08-16

**Authors:** Felix Umrath, Lukas-Frank Schmitt, Sophie-Maria Kliesch, Christine Schille, Jürgen Geis-Gerstorfer, Elina Gurewitsch, Kathleen Bahrini, Fabian Peters, Siegmar Reinert, Dorothea Alexander

**Affiliations:** 1Department of Oral and Maxillofacial Surgery, University Hospital Tübingen, 72076 Tübingen, Germany; felix.umrath@med.uni-tuebingen.de (F.U.); lukas.schmitt3@gmx.de (L.-F.S.); siegmar.reinert@med.uni-tuebingen.de (S.R.); 2Department of Orthopedic Surgery, University Hospital Tübingen, 72076 Tübingen, Germany; 3Quality Analysis GmbH, 72622 Nürtingen, Germany; s.kliesch@qa-group.com; 4Section Medical Materials Science and Technology, University Hospital Tübingen, 72076 Tübingen, Germany; cschille2020@gmail.com (C.S.); geis-gerstorfer@mwt-tuebingen.de (J.G.-G.); 5Curasan AG, 65933 Frankfurt, Germany; elina@gurewitsch.name (E.G.); kathleen.bahrini@curasan.com (K.B.); f.peters@gdfmbh.com (F.P.)

**Keywords:** jaw periosteal cells, mesenchymal stem cells, osteogenic differentiation, biomaterial, scaffold, mechanical properties, Raman spectroscopy, β-tricalcium phosphate, polylactic acid

## Abstract

Autologous bone transplantation is still considered as the gold standard therapeutic option for bone defect repair. The alternative tissue engineering approaches have to combine good hardiness of biomaterials whilst allowing good stem cell functionality. To become more useful for load-bearing applications, mechanical properties of calcium phosphate materials have to be improved. In the present study, we aimed to reduce the brittleness of β-tricalcium phosphate (β-TCP). For this purpose, we used three polymers (PDL-02, -02a, -04) for coatings and compared resulting mechanical and degradation properties as well as their impact on seeded periosteal stem cells. Mechanical properties of coated and uncoated β-TCP scaffolds were analyzed. In addition, degradation kinetics analyses of the polymers employed and of the polymer-coated scaffolds were performed. For bioactivity assessment, the scaffolds were seeded with jaw periosteal cells (JPCs) and cultured under untreated and osteogenic conditions. JPC adhesion/proliferation, gene and protein expression by immunofluorescent staining of embedded scaffolds were analyzed. Raman spectroscopy measurements gave an insight into material properties and cell mineralization. PDL-coated β-TCP scaffolds showed a significantly higher flexural strength in comparison to that of uncoated scaffolds. Degradation kinetics showed considerable differences in pH and electrical conductivity of the three different polymer types, while the core material β-TCP was able to stabilize pH and conductivity. Material differences seemed to have an impact on JPC proliferation and differentiation potential, as reflected by the expression of osteogenic marker genes. A homogenous cell colonialization of coated and uncoated scaffolds was detected. Most interesting from a bone engineer’s point of view, the PDL-04 coating enabled detection of cell matrix mineralization by Raman spectroscopy. This was not feasible with uncoated scaffolds, due to intercalating effects of the β-TCP material and the JPC-formed calcium phosphate. In conclusion, the use of PDL-04 coating improved the mechanical properties of the β-TCP scaffold and promoted cell adhesion and osteogenic differentiation, whilst allowing detection of cell mineralization within the ceramic core material.

## 1. Introduction

The most commonly used scaffold materials for bone tissue engineering are synthetic polymers and ceramics, natural materials, and composites [1]. Examples of polymers used as scaffolds for bone regenerative purposes in dental and craniofacial medicine include polycaprolactone (PCL), polyglycolid (PGA), and polylactic acid (PLA) and their composites poly(lactic-co-glycolic acid) (PLGA) [2]. The degradation rate of these aliphatic polyesters proceeds in the following order: PLGA degrades the fastest, followed by PLA, and PCL degrades the slowest [3]. PLA is a biodegradable and biocompatible polymer that can be processed into various shapes and sizes. It degrades by hydrolysis to lactic acid, a natural metabolite that can be easily eliminated from the body. However, the effect of increasing lactic acid concentration on the decreasing pH in a common DMEM medium containing 10% fetal bovine serum, 1% penicillin–streptomycin and placed in an incubator at 5% CO_2_ has been convincingly demonstrated [4].

Calcium phosphate (CaP)-based ceramics such as hydroxyapatite (HA), β-tricalcium phosphate (β-TCP), calcium polyphosphate (CPP), and biphasic calcium phosphate (BCP) show excellent osteoinductive and osteoconductive properties [5]. β-TCP is chemically similar to hydroxyapatite (HA), the mineral component of natural bone tissue, but exhibits faster biodegradation due to its lower Ca/P ratio compared to hydroxyapatite [6]. β-TCP is a biocompatible ceramic that can be processed into various porous structures, providing a favorable environment for cell attachment and proliferation. However, the low strength and high brittleness of ceramics make them difficult to be used in load-bearing body sites.

The combination of two commonly used materials for bone tissue engineering, PLA and β-TCP, can improve the elastic stiffness and flexural strength of the β-TCP ceramic, and control degradation kinetics while buffering the acidity of lactic acid and promoting new bone tissue formation.

In former studies, we established surface coatings with PLGA to enable the biofunctionalization of inorganic β-TCP blocks with different osteogenic molecules [7] in order to enhance cell functionality within the used ceramic scaffold. In the present study, we aimed at improving the mechanical properties of the β-TCP scaffold and at establishing an approach for the detection of cell matrix mineralization notwithstanding the same inorganic component of the core material. Raman spectroscopy is a useful tool as it can provide non-destructive, label-free and real-time information about the molecular composition and structure of tissues and biomaterials [8]. Furthermore, we are convinced that Raman spectroscopy can be used to assess the quality, maturity and functionality of engineered tissue constructs by detecting changes in the composition and structure of the extracellular matrix, as we were able to demonstrate in previous studies with 2D-cultured JPCs under different media or supplementation conditions [9,10]. The big challenge facing us in the present work was that we wanted to achieve cell mineralization detection despite the intercalating effects of the core ceramic material of β-TCP.

## 2. Materials and Methods

### 2.1. Cell Isolation and Culture of JPCs

JPCs derived from 4 donors were included in this study in accordance with the local ethical committee (approval number 618/2017BO2) and after obtaining written informed consent for the participants.

The jaw periosteal tissue was cut in small pieces with a scalpel and the suspension was cultured in DMEM/F12 + 10% human platelet lysate (hPL provided by the Centre for Clinical Transfusion Medicine in Tübingen, which did not contain heparin and was referred to as a research lysate based on the absent quarantine period). After initial passaging, cells were expanded for up to 4 passages under 5% hPL supplementation until used in passage 5–6 for adhesion, proliferation and osteogenic differentiation experiments. DMEM-cultured cells were passaged using TrypLE Express (Thermo Fisher Scientific, Waltham, MA, USA) and medium change was performed three times per week.

Osteogenic conditions were performed by the addition of 10% hPL, dexamethasone (4 µM), β-glycerophosphate (10 mM) and L-ascorbic acid 2-phosphate (100 µM) for the period indicated in the different sections.

### 2.2. Analyses of Cell Adherence and Viabilities

For the analyses of cell adherence, 5 × 10^4^ cells were seeded per β-TCP scaffold, after the pre-incubation of scaffolds with complete medium for one hour. After 1 h and 3 h of incubation at 37 °C and 5% CO_2_, the residual cell suspension was removed and scaffolds were washed with PBS. Cell suspension was counted (non-adherent cell fraction). Adherent cells were detached from scaffolds by the addition of TrypLE Express (Thermo Fisher Scientific, Waltham, MA, USA). Detached cells were counted and calculated in relation to the total cell number (adherent + non-adherent cell fraction) seeded.

For the comparison of viability of JPCs growing within β-TCP scaffolds with different coatings (PLGA, PDL-02, -02A, -04,), scaffolds were pre-incubated with complete medium for one hour in the incubator, and 5 × 10^4^ cells were seeded per scaffold and incubated in 96-well plates for 8 days. Afterwards, cell-seeded scaffolds were placed in 96-well plates with 200 µL fresh medium and after the addition of 20 µL substrate (EZ4U, Biozol, Eching, Germany) and 4 h of incubation, photometric measurements followed at 450 nm using a microplate reader (Biotek, Friedrichshall, Germany). The same procedure was used for the examination of metabolic activities of JPCs within uncoated and PDL-04 coated β-TCP scaffolds under untreated and osteogenic conditions for 6 and 13 days, respectively.

### 2.3. Scaffold Manufacturing

Pure-phase β-tricalcium phosphate (β-TCP) was synthesized from calcium carbonate and calcium hydrogen phosphate (Merck, Darmstadt, Germany) as described elsewhere [11]. The TCP particles were mixed with ammonium hydrogen carbonate (Merck, Darmstadt, Germany) in particle sizes of 10–500 µm and pressed together using an axial press (Weber PW40). The compacted blocks were set in a drying chamber to remove the NH_4_HCO_3_, leaving in its stead pores of crystal size of the sublimated substance. The compacted, porous substrates were heated at 1100 °C for 8 h. The blocks were then milled using a CNC milling machine (KERN HSPC 2522) into round discs with a diameter of 5 mm and thickness of 3 mm. For the determination of the mechanical properties, cuboids with 45 mm × 6 mm × 6 mm were manufactured.

The scaffolds were impregnated with poly (DL-lactides) PURASORB (Corbion Purac, Gorinchem, The Netherlands) according to Table 1. The polymers are amorphous, from equal R- and S-monomer ratios, have a glass transition temperature of 50–55 °C, a compressing strength of 45–55 MPa and a degradation time of 12–16 months (manufacturer information).

The polymers were dissolved in ethyl acetate to reach a concentration of 20% *w*/*v*. In order to evaluate the PDL-04 polymer uptake efficiency (Table 2), the average weight of uncoated β-TCP scaffolds was determined in a syringe (number of scaffold pieces per syringe is given in the table). After soaking of the scaffolds with petroleum benzine for 2 h, they were placed in the syringe and a PDL-04/ethyl acetate/petroleum benzine solution (2.2:1:1.3) was drawn into the syringe. By pulling and pushing the plunger several times, negative pressure was created in the syringe. Scaffolds were kept in the syringe for 3 days under negative pressure and then dried on ceramic granulates. Thereafter, average weight after coating and the PDL-04 content were determined/calculated. The coating efficiency (*CE*) was calculated following the formula:$$CE\;\left(\%\right)=\frac{PDL\;content\;\left(g\right)}{weight\;after\;coating\;\left(g\right)}\times 100(\%)$$

Further, we evaluated the water absorption capacity of PDL-coated β-TCP scaffolds (Table 3). After the determination of the dry weight of PDL-02/PDL-04 coated β-TCP scaffolds, samples (*n* = 4 per group) were rinsed in VE water for 90 s and wet weight was determined. The water content was calculated by the subtraction of the dry from the wet weight. The water absorption capacity (*WAC*) was calculated by the formula:$$WAC\;\left(\%\right)=\frac{water\;content\;(g)}{wet\;weight\;(g)}\times 100\;(\%)$$

The pure and coated β-TCP scaffolds were packed in Cleerpeel^®^ Pouches (Sengewald/Coveris, Rohrdorf, Bavaria, Germany) and sterilized using gamma radiation.

### 2.4. Degradation Kinetics

After gamma sterilization, 0.5 g of each polymer sample was immersed in 20 mL of deionized water. In addition, three polymer-coated β-TCP scaffolds of each sample were treated with water. Pure water was used as a reference. The samples were incubated in a shaking incubator at 37 °C (GFL, 3032, LAUDA, Lauda-Königshofen, Germany) for 6 weeks. At regular intervals, pH and electrical conductivity were measured using a WTW Multiline P3 pH meter (WTW, Weilheim, Germany).

### 2.5. SEM Examination

Polymers and polymer-coated scaffolds were examined using an AMRAY 1810 p (Amray Inc., Bedford, MA, USA) scanning electron microscope. The samples were sputtered with graphite and gold to prevent electrical charging of the calcium phosphate and polymer.

### 2.6. Mechanical Properties of Uncoated and PLA-Coated β-TCP Scaffolds

For the strength test, β-TCP samples measuring 6 × 6 × 45 mm were supplied by Curasan AG in a gamma-sterilized condition and were sealed in plastic bags. The three PLA coating variants with the designations PDL-02, PDL-02A and PDL-04 were examined (*n* = 8) in comparison with the uncoated β-TCP material as a control (*n* = 6). The dimensions of the test specimens were measured with a digital caliper gauge with a reading accuracy of 0.05 mm at three points in the material and results were averaged.

The mechanical parameters determined in the delivered materials were the bending strength σ, the modulus of elasticity *E* and the bending elongation at break *A*_g_.

The 3-point flexure test was used (Figure 1) to evaluate the mechanical properties (support span 40 mm, crosshead speed 0.5 mm/min, Zwick Z010 testing apparatus with the “Test Expert” version 12 software).

The 3-point bending strength was calculated using the formula:$$\sigma =\frac{3PI}{{2wb}^{2}}$$

*P* is the breaking load, in Newtons; *I* is the test span (center-to-center distance between support rollers), in millimeters; *w* is the width and *b* is the thickness of the specimen in millimeters. Eight test specimens were measured for each coating variant and six for the uncoated control. The respective mean value and standard deviation were calculated from the measurement results. After the bending test, the fracture surfaces of each specimen were photographed with a stereo microscope (M400, Wild, Heerbrugg, Switzerland) at a 20× magnification to identify any internal defects.

### 2.7. Histological Examination of Cell-Seeded Scaffolds

The cell-seeded uncoated and PDL-04 coated β-TCP scaffolds were embedded in Technovit^®^ 9100 (on PMMA basis, Kulzer GmbH, Hanau, Germany) according to modified manufacturer’s instructions. Microtome sections were prepared by cutting the embedded scaffolds using a fully electric rotary microtome (pfm Rotary 3006 EM; pfm medical titanium GmbH, Nuremberg, Germany) with a hard-cutting blade (SH35W Feather^®^ microtome blade HPTC; Feather Safety Razor Co., LTD., Osaka, Japan). The thickness of the sections was 5 μm. The sample was first cut in half with a diamond saw to expose the cross-section. The resulting sections were mounted on glass slides (SuperFrost; Thermo Fisher Scientific, Waltham, MA, USA).

### 2.8. Toluidine Blue Staining

First, the sections and slices were etched with 10% acetic acid for 5 min. The sample was then washed with distilled water and dried. In the next step, a drop of toluidine staining solution (0.5 mg/mL) was applied to the section with a syringe (B. Braun AG, Melsungen, Germany) for 15 min. Sterilization filtration with a 0.2 μm membrane filter (Sartorius, Göttingen, Germany) was used to ensure that no contamination or small dye particles were present in the staining solution. The staining solution was then briefly washed with water. DePeX was used to cover the slides.

### 2.9. Mineralization Detection with OsteoImage

To examine mineralization, the microtome sections were first deacrylated in 100% xylene for 4 min and then stained with the OsteoImage™ Mineralization Assay Kit (Lonza, Walkersville, MD, USA). Hoechst33342 (Promo-Kine, Heidelberg, Germany) was used for nuclear staining. After washing and drying, the sections were covered with glycerin. Fluorescence intensity of OsteoImage staining was quantified using ImageJ software and normalized to the intensity of the nuclear staining.

### 2.10. Immunofluorescence Staining

Prior to staining, the microtome sections were deplasted in 100% xylene for one minute. Next, the cells were permeabilized with 0.1% Triton X-100 for 10 min. To prevent nonspecific antibody binding, the sections were then blocked with 2% bovine serum albumin (BSA) for 1 h and washed with PBS for 5 min. The primary antibody (anti-OCN, monoclonal mouse antibody, isotype: IgG1, MAB1419; R&D Systems, Minneapolis, MN, USA) was diluted to a concentration of 10 μg/mL in 2% BSA and stored overnight at 4 °C in a humid chamber. The next day, the sections were washed with PBS, and a DyLight405-labeled secondary antibody (polyclonal goat anti-mouse IgG1 antibody, 409109; Biolegend, San Diego, CA, USA) was diluted to a concentration of 5 μg/mL in 2% BSA and incubated for one hour. The sections were then washed with PBS, and a nuclear staining solution with Sytox Orange (5 mM stock solution diluted 1:1000 in PBS, ThermoFisher Scientific, Waltham, MA, USA) was incubated on the slides for 10 min. Fluorescence intensity of ALP and OCN immunostaining was quantified using ImageJ software and normalized to the intensity of the nuclear staining.

### 2.11. Gene Expression Analyses by Quantitative PCR

Cell-seeded scaffolds were placed in Lysing Matrix D microtubes containing ceramic beads (MP Biomedicals, Irvine, CA, USA) and lysis buffer (Macherey-Nagel, Dueren, Germany), and shredded by a FastPrep-24 device (MP Biomedicals, Irvine, CA, USA). RNA isolation from JPCs growing within uncoated or polymer-coated β-TCP scaffolds was performed using the NucleoSpin RNA kit (Macherey-Nagel, Düren, Germany) following the manufacturer’s instructions. RNA concentration was measured using a Qubit 3.0 fluorometer and the corresponding RNA BR Assay Kit (Thermo Fisher Scientific Inc., Waltham, MA, USA). A total amount of 0.5 μg RNA was used for the first-strand cDNA synthesis using the SuperScript Vilo Kit (Thermo Fisher Scientific Inc., Waltham, MA, USA). The quantification of mRNA expression levels was performed using the real-time LightCycler System (Roche Diagnostics, Mannheim, Germany). For the PCR reactions, commercial primer kits (Search LC, Heidelberg, Germany), and DNA Master SYBR Green I (Roche, Basel, Switzerland) were used. The amplification of cDNAs was performed with a touchdown PCR protocol of 40 cycles (annealing temperature between 68 and 58 °C), following the manufacturer’s instructions. Copy numbers of each sample were calculated on the basis of a standard curve (standard included in the primer kits) and normalized to the housekeeping gene glyceraldehyde-3-phosphate dehydrogenase (GAPDH).

### 2.12. Raman Spectroscopy

An inVia Qontor Raman microscope (Renishaw GmbH, Pliezhausen, Germany) was employed for all measurements at Quality Analysis GmbH (Nürtingen, Germany). Raman spectra were excited by a 785 nm laser beam through a 40× water-immersion objective (Leica Microsystems GmbH, Wetzlar, Germany). The system was calibrated based on the silicon peak at 520 cm^−1^ prior to all measurements. The laser output power was set to 150 mW (due to scattering losses, only 50% of the laser power reaches the sample) for spectra acquisition. Raman spectra were collected from either cell-free scaffolds (β-TCP or β-TCP_PDL-04), or from scaffolds seeded with JPCs and cultured under control (CO) or osteogenic (OB) conditions for 17 days. All scaffolds were transferred for Raman measurements into uncoated glass bottom cell culture dishes (Greiner Bio-One GmbH, Frickenhausen, Germany), henceforth referred to as Raman dishes. The total acquisition time per spectrum was 3 s within a mapping measurement with 90–148 single spectra. A background spectrum from the glass dish containing the cell culture medium was taken for each set of data. All acquired Raman spectra were background-subtracted using the specific background spectrum and then baseline corrected (arithmetic operation) using WiRE (Renishaw GmbH, Pliezhausen, Germany). A smoothing algorithm (Savitzky-Golay, second polynomial order, 7 data points) was employed on all spectra.

Raman spectra were employed to compare the biochemical composition of scaffolds and JPC-formed extracellular matrix. For this purpose, the following peaks and ratios were analyzed: PLA (874 cm^−1^), P–O (PO_4_^3−^, 971 cm^−1^), νP–O (1091 cm^−1^), P=O (1374 cm^−1^), HA/amide III (960/1244 cm^−1^) carbonate/HA (1070/960 cm^−1^). For the calculation of HA crystallinity (crystal size), the inverse of the full-width half maximum (FWHM) was calculated for the spectral range from 900–1000 cm^−1^ using WiRE software. The apatite peak was fitted using a standard Gaussian curve. Based on the fitted curve, the inverse of FWHM (1/FWHM) was calculated for the HA peak at 960 cm^−1^.

### 2.13. Statistical Analysis

For the statistical evaluation of the mechanical parameters, one-way ANOVA and Tuckey’s multiple comparison test were performed with all coating variants versus the uncoated controls. Gene expression and metabolic activity values are displayed as means ± SD and were compared using one-way ANOVA with Tuckey’s multiple comparison test. Adhesion, proliferation and alkaline phosphatase expression of JPCs growing on PDL-04-coated and uncoated β-TCP scaffolds were compared using two-way ANOVA with Šídák’s multiple comparisons test. Peak intensities and ratios of normalized Raman spectra were compared using unpaired *t*-tests or two-way ANOVA. A *p*-value < 0.05 was considered significant.

## 3. Results

### 3.1. Polymer Coating Efficiency/Water Absorption Capacity

The polymer uptake of the β-TCP material was 10% with PDL-02 and PDL-02a. The polymer uptake with PDL-04 was initially 12% and 14.53 ± 3.13% after further optimization (Table 2).

**Table 2 jfb-14-00427-t002:** PDL-04 coating efficiency of β-TCP scaffolds. After the determination of the average weight of scaffolds before and after PDL-04 coating, the PDL-04 content and the mean coating efficiency [%] were calculated.

Experimental Series	Reaction Device (Syringe)	Number of Scaffolds per Syringe [Piece]	Average Weight before Coating [g]	Average Weight after Coating [g]	PDL-04 Content [g]	Coating Efficiency [%]	Mean PDL-04 Coating Efficiency [%]
#1	1	50	3.157	3.684	0.527	14.3	14.53 ± 3.13
	2	50	3.285	3.716	0.431	11.6	
	3	55	3.605	4.166	0.561	13.5	
	4	54	3.533	3.761	0.228	6.1	
#2	1	50	3.283	3.894	0.611	15.7	
	2	50	3.269	3.972	0.703	17.7	
	3	55	3.613	4.487	0.874	19.5	
#3	1	56	3.662	4.236	0.574	13.6	
	2	55	3.592	4.304	0.712	16.5	
	3	56	3.644	4.262	0.618	14.5	
	4	56	3.646	4.434	0.788	17.8	
	5	56	3.657	4.378	0.721	16.5	
	6	55	3.592	4.112	0.520	12.6	
	7	55	3.579	4.271	0.692	16.2	
	8	55	3.587	4.120	0.533	12.9	
	9	53	3.466	4.003	0.537	13.4	

Results from 3 experimental series for a total of 808 specimens are given in Table 2.

Due to the concentration of the dripping solutions, the materials were still porous. Water-soaking experiments showed a continuous capillarity due to micropores. The calculated water absorption capacity of PDL-02 and PDL-04 coated β-TCP scaffolds is given in Table 3. WAC values for PDL-02A coated β-TCP scaffolds could not be evaluated as they disintegrated quickly in VE water. The mean WAC of PDL-04 coated β-TCP scaffolds (14.4 ± 1.48%) was shown to be significantly lower (* *p* < 0.05) than that of PDL-02 coated composites (24.7 ± 5.6%).

**Table 3 jfb-14-00427-t003:** Evaluation of the water absorption capacity of PDL-coated β-TCP scaffolds. Dry weight [g], wet weight [g], water content [g], water absorption capacity [%], and mean WAC [%] of PDL-02- and PDL-04-coated β-TCP scaffolds are given in the table (* *p* < 0.05).

Sample	Dry Weight (g)	Wet Weight (g)	Water Content (g)	Water Absorption Capacity (WAC, %)	Mean WAC (%)
PDL-02-1	0.291	0.384	0.094	24.4	24.7
PDL-02-2	0.283	0.376	0.093	24.8	
PDL-02-3	0.283	0.414	0.131	31.7	
PDL-02-4	0.283	0.345	0.062	18.0	
PDL-04-1	0.290	0.334	0.044	13.1	14.4 *
PDL-04-2	0.285	0.328	0.043	13.1	
PDL-04-3	0.288	0.343	0.054	15.8	
PDL-04-4	0.289	0.342	0.053	15.5	

### 3.2. Degradation Kinetic Measurements

The degradation of the polymers and polymer-coated scaffolds was monitored by pH (Figure 2A) and electrical conductivity measurements (Figure 2B). The pure polymers degrade fast after gamma-sterilization in the first 40 days. PDL-02 and PDL-02A degrade faster and show a burst increase of electrical conductivity. Possibly, the autocatalytic reaction leads to a fast accumulation of ionic lactic acid in these samples. PDL-04 degrades more slowly and has a rather moderate increase in electrical conductivity of the surrounding fluid.

The slower degradation of the polymer-coated β-TCP scaffolds is caused by the buffering capacity of the calcium phosphate (Figure 3). Lactic acid degradation products dissolve TCP. Therefore, the pH values remain higher compared to the pure polymers. The electrical conductivity of the PDL-02 and PDL-02A coated samples shows a slower and rather moderate increase (Figure 3B). The scaffolds coated with PDL-04 show no significant change in the pH value and a very slow increase in electrical conductivity indicating a very slow polymer/scaffold degradation (Figure 3A).

### 3.3. SEM Examination of the Degraded Samples

SEM examination of gamma-irradiated polymer samples after 180 days’ degradation showed different results. The PDL-02 (Figure 4A) remains nearly stable and shows surface degradation characteristics. The PDL-02A polymer decomposed into smaller particles (Figure 4B). The PDL-04 remained stable and showed degradation and crystallization on the surface (Figure 4C,D).

The SEM images of the degradation of the coated scaffolds reflected the degradation kinetics measured by pH and electrical conductivity. The PDL-02 coated scaffolds showed particular decomposition due to the low pH level (Figure 5A). Also, PDL-02A coated samples show surface leaching, visible by microcracks and degradation lacunae (Figure 5B). The surface of the PDL-04 coated substrates remains stable, and only a few microcracks on the sinternecks are visible, showing moderate degradation and dissolution of the β-TCP particles (Figure 5C,D). Figure 5E shows the overview of 3 β-TCP scaffolds clearly demonstrating that PDL-04 coated composites are the most stable ones.

### 3.4. Measurements of the Mechanical Properties of Coated Compared to Uncoated β-TCP Constructs

The results of the mechanical testing of the coated β-TCP variants in comparison to the uncoated control are shown in Figure 6A–C. Differences between uncoated and coated scaffolds were highly significant. Further, significant differences were detected between the PDL-02 and PDL-04 coating variants.

The results of the mean 3-point flexural strength σ were 4.92 ± 0.37 MPa for PDL-02, 5.19 ± 0.27 MPa for PDL-02A and 5.68 ± 0.88 MPa for PDL-04, while the mean value of the uncoated control was significantly lower at 0.47 ± 0.07 MPa (Figure 6A). A significant difference was also detected between the PDL-02 and PDL-04 coated scaffolds.

By the analysis of the modulus of elasticity *E*, we obtained 820.8 ± 62.4 MPa for PDL-02, 864.3 ± 44.8 MPa for PDL-02A, and 947.4 ± 146.9 MPa for PDL-04, all more than 10-fold higher values compared to 78.6 ± 11.9 MPa for the uncoated controls (Figure 6B).

The mean values of bending elongation at break *A*_g_ were 0.12 ± 0.01 mm for PDL-02, 0.12 ± 0.01 mm for PDL-02A, 0.13 ± 0.02 mm for PDL-04, and 0.05 ± 0.0 mm for the uncoated control (Figure 6C).

The results demonstrate that the three PLA-coatings provide a significant improvement in mechanical strength values of β-TCP scaffolds by a factor of about 10 compared to the uncoated control. In contrast, the differences in flexural strength and modulus of elasticity between the three analyzed coatings were rather small and only significantly higher in β-TCP samples coated with PDL-04 compared to values detected for PDL-02 coated scaffolds.

### 3.5. Proliferation Activities of JPCs Growing within Different Composites

In order to decide which coating variant is the best in terms of cytocompatibility and functionality, we compared metabolic activity and gene expression of JPCs growing within the four coated scaffold variants (PDL-02, PDL-02A, PDL-04, PLGA—internal control). Since metabolic activity is proportional to the number of cells present in a scaffold, it was used here to measure cell proliferation activities.

JPCs growing within β-TCP scaffolds coated with PDL-02A showed the lowest proliferation activities compared to the other three groups at day 8 of cell culture. The highest proliferation activities were detected in PDL-04 and PLGA coated scaffolds, both with significant differences (*p* < 0.05) compared to the PDL-02A variant (Figure 7).

### 3.6. Gene Expression in JPCs Growing within Different β-TCP Composites

Gene expression analyses (Figure 8) of osteogenesis-related genes (alkaline phosphatase (ALPL), runt-related transcription factor-2 (RUNX2), osteoprotegerin (OPG) and type I collagen (COL1A1)) showed a tendency (without reaching significant differences) for highest mRNA levels in JPCs growing within the variant PDL-04 type. The lowest levels of ALPL, RUNX2, OPG and COL1A1 gene expression were detected in JPCs growing within PDL-02 and PLGA. The highest mRNA expression levels for the osteogenesis relevant genes were detected in JPCs growing within the PDL-04, albeit without reaching significant differences.

Since gene expression levels of the analyzed genes showed a tendency for highest levels in JPCs growing within PDL-04, we decided to choose this coating variant for further analyses and comparison to uncoated β-TCP scaffolds.

### 3.7. Cell Attachment, Proliferation and Alkaline Phosphatase Activity of JPCs Growing on Uncoated and PDL-04 Coated Scaffolds

As shown in Figure 9A, significantly higher rates of adherent cells were detected in PDL-04 coated compared to uncoated β-TCP scaffolds (51.1% vs. 36.3% and 59.6% vs. 31.8%) at both 1 h and 3 h after cell seeding.

Despite lower initial adherence, JPCs showed higher proliferation rates in control medium on uncoated scaffolds at day 6 and 13 (Figure 9B). In osteogenic medium, no differences in proliferation activities between coated and uncoated JPC-seeded β-TCP scaffolds were observed.

As shown in Figure 9C, significantly higher ALP activities were detected in JPCs growing on PLA-coated compared to uncoated scaffolds, when cells were cultivated in osteogenic medium.

### 3.8. Histological Examination of JPC-Seeded Uncoated and Coated (PDL-04) Scaffolds

Histologic examination by toluidine blue staining revealed a thin but very dense cell layer within PDL-04 coated scaffolds (Figure 10A,C), whereas JPCs were loosely distributed within a thicker layer on the surface of uncoated scaffolds (Figure 10B,D).

In order to detect hydroxyapatite formation in JPCs growing within PDL-04 coated and uncoated scaffolds, 5 µm sections of embedded scaffolds were stained by OsteoImage and counter stained with Hoechst 33342, as shown in Figure 11.

For the negative control illustrated in Figure 11A, a cell-free coated scaffold was used. For the positive control, human fibula (Figure 11B) was stained. As shown in Figure 11C,D, JPCs growing within uncoated β-TCP scaffolds did not produce hydroxyapatite. Only JPCs growing on PDL-04 coated β-TCP scaffolds showed colocalization of Hoechst and OsteoImage under osteogenic conditions (Figure 11F) whereas the undifferentiated control (Figure 11E) remained negative. The quantification diagram shown in Figure 11G reflects the calculated significances for the OsteoImage staining.

Detection of alkaline phosphatase expression is shown in Figure 12. Similarly to how we could demonstrate for the OsteoImage staining, colocalization of blue fluorescence (anti-ALP primary and DyLight secondary antibodies) and red fluorescence for nuclear staining was detected only in the positive control (human fibula, Figure 12B) and in PDL-04 coated JPC-seeded β-TCP scaffolds which were osteogenically induced for 17 days (Figure 12F). The quantification diagram shown in Figure 12G reflects the differences in ALP expression between analyzed samples, albeit without reaching significant differences.

Detection of osteocalcin expression as a late marker for cell mineralization is shown in Figure 13. Colocalization of blue fluorescence (anti-OCN and DyLight antibodies) and red fluorescence for nuclear staining was detected in the positive control (human fibula, Figure 13B) and in PDL-04 coated JPC-seeded β-TCP scaffolds which were osteogenically induced for 17 days (Figure 13F). A weak staining was also observed in uncoated osteogenically induced JPC-seeded β-TCP scaffolds (Figure 13D). Figure 13G depicts the calculated significant differences.

### 3.9. Raman Spectroscopy

Uncoated and PLA-coated scaffolds seeded with JPCs were analyzed by Raman spectroscopy in order to detect differences in the quality of matrix mineralization under osteogenic (OB) compared to untreated control (CO) conditions. Figure 14 illustrates representative Raman spectra of cell-free PDL-04 coated and uncoated β-TCP scaffolds, which were incubated with control and osteogenic medium for 17 days.

The spectra of the coated scaffolds showed a distinct peak at 874 cm^−1^ (red arrow) which is specific for the PLA coating. Further, higher phosphate peaks were detected when scaffolds were treated with osteogenic (OB) medium compared to control (CO) medium. However, uncoated scaffolds show higher phosphate peaks compared to PLA-coated scaffolds under the same conditions. This is probably due to masking of the phosphate signals by the PLA coating. As shown in Figure 15, differences in phosphate peak intensities were highly significant between coated and uncoated cell-free scaffolds.

To analyze changes in ECM deposition and matrix mineralization by JPCs under osteogenic and control conditions, different hydroxyapatite (HA) to protein ratios (HA to phenylalanine (Phe), HA to amide III) were calculated from the mean Raman spectra based on the corresponding peak intensities (as shown in Figure 16).

As illustrated for PLA-coated scaffolds (*p* < 0.0001), osteogenic differentiation leads to higher HA intensities and therefore to a higher HA/Phe ratio in OB samples (Figure 16A). Interestingly, uncoated scaffolds show the opposite result, with a significantly (*p* < 0.0001) higher HA/Phe ratio in CO samples. The same result was observed for HA/amide III ratio, although without significant differences between CO and OB samples of PLA-coated scaffolds (Figure 16B).

Furthermore, the carbonate to phosphate ratio (carbonate to HA, Figure 16C) was assessed. As carbonate is a scaffold compound, the ratio should decrease with increase in HA formed by seeded JPCs after osteogenic induction. On uncoated scaffolds, no significant differences between CO and OB samples could be detected. Coated scaffolds showed a significantly lower carbonate/HA ratio when cultured under osteogenic conditions indicating HA deposition during matrix mineralization.

Mineral crystallinity was analyzed by calculating the inverse FWHM (1/FWHM) of the HA peak at 960 cm^−1^ (Figure 16D). No differences between CO and OB samples could be detected on uncoated scaffolds. In contrast, significantly higher 1/FWHM values were detected in cell-seeded PLA-coated scaffolds treated with OB medium compared to the untreated control group.

## 4. Discussion

In the present study, mechanical as well as degradation properties of β-tricalcium phosphate (β-TCP) scaffolds coated with different types of polylactic acid (PLA) were investigated. Secondly, Raman spectroscopy was used in order to establish whether this technology is able to assess matrix mineralization by JPCs growing within the β-TCP scaffolds.

In the dental sector, both flexural strength and compressive strength are important considerations for bone substitute materials. We chose the 3-point flexural test because flexural strength (and especially elasticity) testing is usually more sensitive than compressive strength testing in evaluating the performance of bone graft substitutes for certain functional dental applications when considering the mastication process, which are frequently subjected to flexural forces. Several approaches have been explored to improve the mechanical properties of β-TCP for its potential use in bone replacement and tissue engineering applications. Some of the techniques and strategies include incorporating (doping) other materials, such as hydroxyapatite (HA) or bioactive glass; creating composites by combining β-TCP with polymers or other ceramics; controlling the sintering process and applying heat treatment to optimize the crystal structure and densification of β-TCP; and 3D printing or additive manufacturing techniques of porous structures with tailored scaffold architecture to enhance mechanical performance while promoting tissue ingrowth. Reinforcements by incorporating fibers, such as carbon or polymer fibers, into β-TCP scaffolds can provide additional support. Nanoengineering of β-TCP can lead to enhanced mechanical properties due to increased surface area and altered crystal structure, as exemplified in a recent review article of Li and co-authors [12]. Despite the progress made in improving the mechanical properties of β-TCP, several challenges and pitfalls remain: β-TCP is inherently brittle, which can limit its use in load-bearing applications. Moreover, while β-TCP is bioresorbable, its degradation rate may not always match the pace of bone regeneration, leading to potential issues in the long-term stability of the implant. Achieving strong and stable interfaces between β-TCP scaffolds and host bone is crucial for successful integration.

Due to their brittleness, CaP scaffolds have been limited to non-load-bearing applications. Studies of biphasic CaP (BCP) composites consisting of HA and β-TCP showed a decrease in strength with increasing β-TCP content. This means that the benefit of improved bioactivity by the addition of β-TCP does not correlate with an improvement in strength [13]. On the other hand, while HA increases osteoconductivity for some polymers [14], its addition does not seem to have a positive impact on the strength of polymer-based composites [13]. The compressive strengths from polymer/HA composites fall below that of cancellous bone. Due to this fact and considering the degradation behavior of polymers, it can be assumed that the strength in vivo may decrease prior to increasing. An alternative to increasing CaP material toughness may be to incorporate polymer into the CaP scaffolds rather than incorporating CaP particles into polymer matrices. This was the strategy we used in our study reaching a 5-fold increase of the 3-point flexural strength and a 900-fold increase in the modulus of elasticity (Figure 6A,B).

Another parameter that determines material strength is the pore size and pore fraction. Most previous and some recent studies are of the general opinion that scaffolds with macropores (mostly >100 µm) should be preferentially used for bone tissue engineering purposes since they mimic the structure of trabecular bone. However, macropores weaken the mechanical properties of the brittle CaP materials whereas micropores have the opposite effect and improve their strength [15]. The β-TCP material used in the present study usually contains micro-, meso- and macropores in the range of 50–500 μm, and for instance PLGA coating did not change the mean pore size, as we described previously [7]. Failure to measure the pore size is thus a limitation of the present study. Nevertheless, the calculation of the water absorption capacity given in Table 3 indicates the presence of pores albeit to a lesser extent in PDL-04 compared to PDL-02 coated composites. In the aforementioned previous study from our lab [7], PLGA coating generated a hydrophobic surface which did not significantly differ from the hydrophobic uncoated surface. Despite the fact that hydrophobic surfaces are supposed not to be attractive for cells, we cannot confirm this point either in the present work nor in previous studies with β-TCP constructs.

The enantiomeric forms of the PLA polymer are poly-D-lactic acid (PDLA) and poly-L-lactic acid (PLLA). The modulus of elasticity of PDLA was described to be in the range of 1.4–2.8 GPa and that of PLGA to be quite similar, between 1.4–2.08 GPa [16]. We used PDLA in our study in order to enhance the mechanical resistance of β-TCP, taking into consideration that PDLA degrades more slowly compared to PLGA.

The outer cortical bone layer has been described as having a mechanical resistance in the range of 12–18 GPa, that of the inner cancellous bone, which has a high porosity coefficient of about 80–90%, between 0.1 and 0.5 GPa [16]. In the review article of Dec and co-authors, slightly different values can be found: for cancellous bone in the range of 0.1–2 GPa and for cortical bone of 5–20 GPa [3]. In the present study, PDL-04 coating led to a significant improvement of 3-point flexural strength σ, modulus of elasticity *E*, and bending elongation at break *A*_g_ values, compared to uncoated scaffolds. Further, we detected significant differences of 3-point flexural strength and the modulus of elasticity between the PDL-04 and PDL-02 coatings. Comparing the different experimental groups, with the PDL-04 coating, we were able to reach the highest average mechanical strength, yielding values of 0.9 GPa which lies between those detected for cancellous and cortical bone, but still remaining far from the mechanical properties of compact bone. However, the theory is that implanted pre-maturated tissue engineering constructs should further gradually improve in mechanical functionality within the bone defect site.

PDL-02 and PDL-02A polymers degraded much faster than PDL-04, which was demonstrated by pH and conductivity measurements in deionized water over 180 days. When used for scaffold coating, PDL degradation was slowed down due to the buffering function of the core material β-TCP. Further, pH and conductivity measurements over the same time period, as well as SEM imaging, showed the fastest degradation of PDL-02A coated scaffolds, followed by PDL-02, while PDL-04 coated scaffolds remained relatively stable over the examined time period. This finding is also clearly visible macroscopically in the overview images of PDL-02, PDL-02A and PDL-04 coated composites in Figure 5E.

JPC’s mitochondrial activities corresponded to the observed degradation behavior indicating a negative effect of degradation products on cell proliferation. The significant decrease in JPC mitochondrial activity detected within PDL-02A compared to PDL-04 coated scaffolds is based with a high probability on a change in pH caused by its fast degradation in aqueous solution. Acidosis has been described as inducing a G1-arrest of the cell cycle and a strong increase in necrotic cell death, but not in apoptosis. The mitochondrial oxygen consumption increased gradually with decreasing pH, as described in a study of Rauscher and co-authors [17]. For our 3D cell culture, we used the DMEM/F-12, GlutaMAX™ medium, the underlying formulation of which contains 29 mM sodium bicarbonate which should equilibrate to pH 7.4 in 5% CO_2_. Whether the polymer degradation behavior under cell culture conditions is comparable to that measured in deionized water is uncertain and unlikely. Unfortunately, we did not perform pH measurements within the 3D cell culture dishes during the incubation time indicating another limitation of our study. Nevertheless, the comparatively fast PDL-02A polymer degradation seemed to have an impact on the detected decrease in JPC viability. Conversely, the slow degradation behavior of PDL-04 seemed to have not only beneficial effects on cell proliferation but also on JPC differentiation potential as detected by gene expression of osteogenic markers as well as by immunohistological staining detecting colocalization of alkaline phosphatase and osteocalcin. Similar findings are reported by other studies where the proliferation of mesenchymal stem cells (MSCs) and expression of matrix proteins were pH susceptible [18]. Local pH in the microenvironment of tissue-engineered constructs seems to be crucial for new bone formation, as reported by Monfoulet and colleagues [19]. The researchers could demonstrate that formation of mineralized nodules in the extracellular matrix of hBMSC was fully inhibited at alkaline (>7.54) pH values and that a pH range (specifically, 7.9–8.27) inhibited the osteogenic differentiation of these cells.

In the current publication, we were also able to achieve the second goal of establishing Raman spectroscopy for the identification of cell-formed hydroxyapatite particles within a core material of the same chemical composition. As shown in Figure 14 and Figure 15, Raman measurements identified a characteristic peak for the PLA coating at 874 cm^−1^. This PLA-specific peak was detected also by other researchers [20,21]. At the same time, polymer coating seemed to cover the typical phosphate and HA peaks of the core material. As expected, Raman spectra illustrated in Figure 14 show clearly that detected phosphate peaks are completely the same in all analyzed samples. However, the height of measured peaks differs: the lowest height was detected in JPC scaffolds under undifferentiated conditions while maximal levels were found under osteogenic conditions due to HA formation by the cells and resulting increase in phosphate. On the other hand, lower phosphate peaks are detected in PLA-coated than in uncoated scaffolds. This observation gives the impression that HA formation by the cells is not possible to detect by Raman within the β-TCP material. This is true for uncoated scaffolds. However, calculated spectral ratios of HA/phenylalanine, HA/amide III, carbonate/HA and inverse of FWHM show clearly the expected picture compared to untreated controls. By the increase in cell-formed HA induced by osteogenic conditions of 3D-cultured JPCs, we expect higher ratios of HA/phenylalanine, HA/amide III and inverse of FWHM values and lower ratios of carbonate/HA, exactly as shown in Figure 15. In our study, higher mineral crystallinity seemed to correlate with lower carbonate to phosphate ratios. This correlation was also reported by others [22]. We conclude that masking of material-specific HA signal by PLA coating enables the detection of cell-formed HA within the used β-TCP material.

## 5. Conclusions

In conclusion, the combination of the herein used PLA material and β-tricalcium phosphate (β-TCP) exhibits synergistic effects for the development of functional bone tissue engineering constructs. The conducted polymer coating significantly enhanced the mechanical properties of used β-TCP scaffolds, and its low degradation kinetics created a favorable microenvironment for the functionality of JPCs. Secondly, PLA coating enabled the detection of cell mineralization by masking material-specific calcium phosphate, highlighting the suitability of the Raman technology to monitor the maturity grade of PLA/β-TCP constructs.

## Figures and Tables

**Figure 1 jfb-14-00427-f001:**
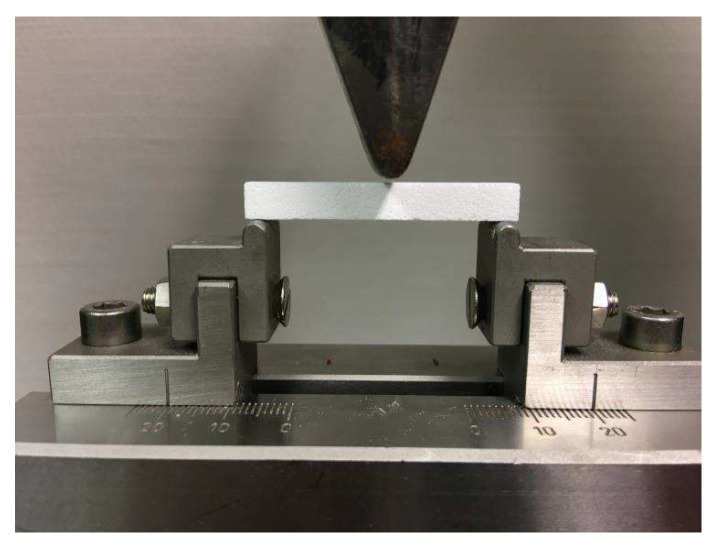
Measurement device for the 3-point flexure test. The support span was 40 mm, the crosshead speed 0.5 mm/min, device: Zwick Z010 (Zwick Roell, ULM, Germany) with the “Test expert” version 12 software.

**Figure 2 jfb-14-00427-f002:**
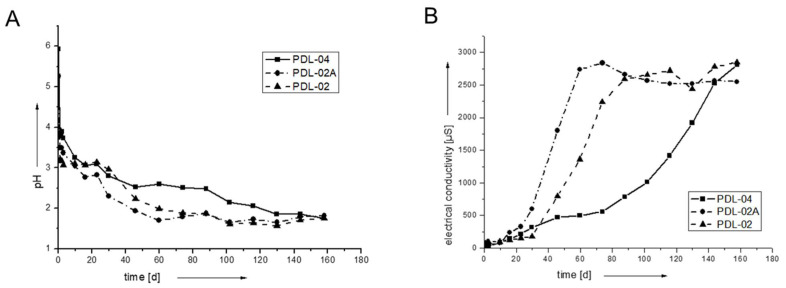
pH (**A**) and conductivity (**B**) of three resorbable PLA-polymers (PDL-02, PDL-02A, PDL-04). Over time, the pH values decrease to below pH 2 and conductivity increases due to proceeding degradation and release of lactic acid. PDL-04 showed the slowest degradation as reflected by higher pH and lower values for electrical conductivity.

**Figure 3 jfb-14-00427-f003:**
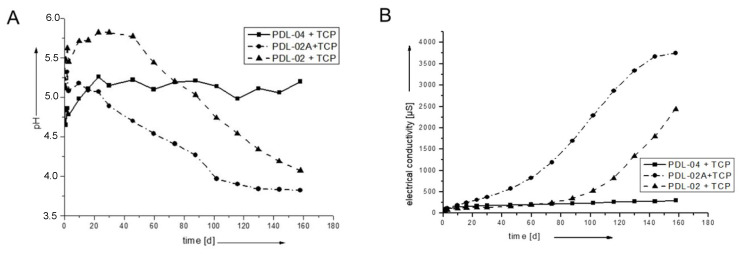
pH (**A**) and conductivity (**B**) of Ø 5 × 3 mm β-TCP scaffolds, coated with the three polymers. The buffering capacity of TCP is clearly visible. The degradation of PDL-04 is the most moderate and stable.

**Figure 4 jfb-14-00427-f004:**
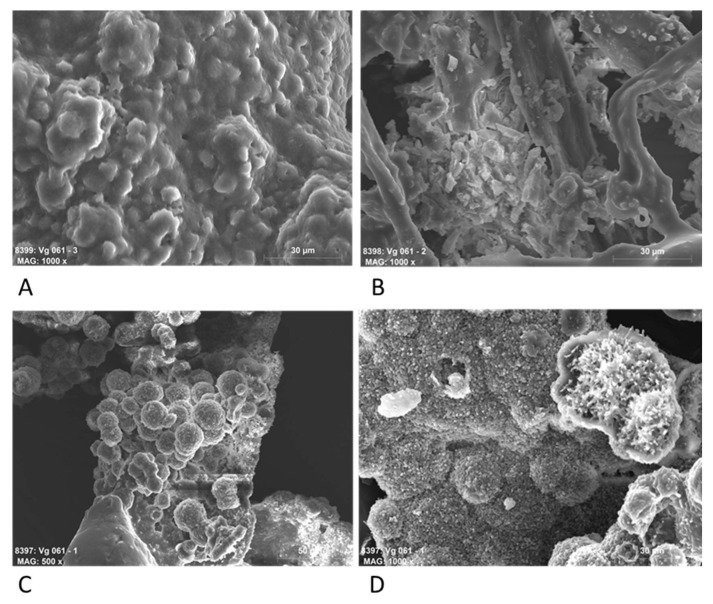
SEM photos of the polymers after 180 days of degradation in deionized water. (**A**) PDL-02 (magnification 1000×); (**B**) PDL-02A (1000×); (**C**,**D**) PDL-04 (500× and 1000×).

**Figure 5 jfb-14-00427-f005:**
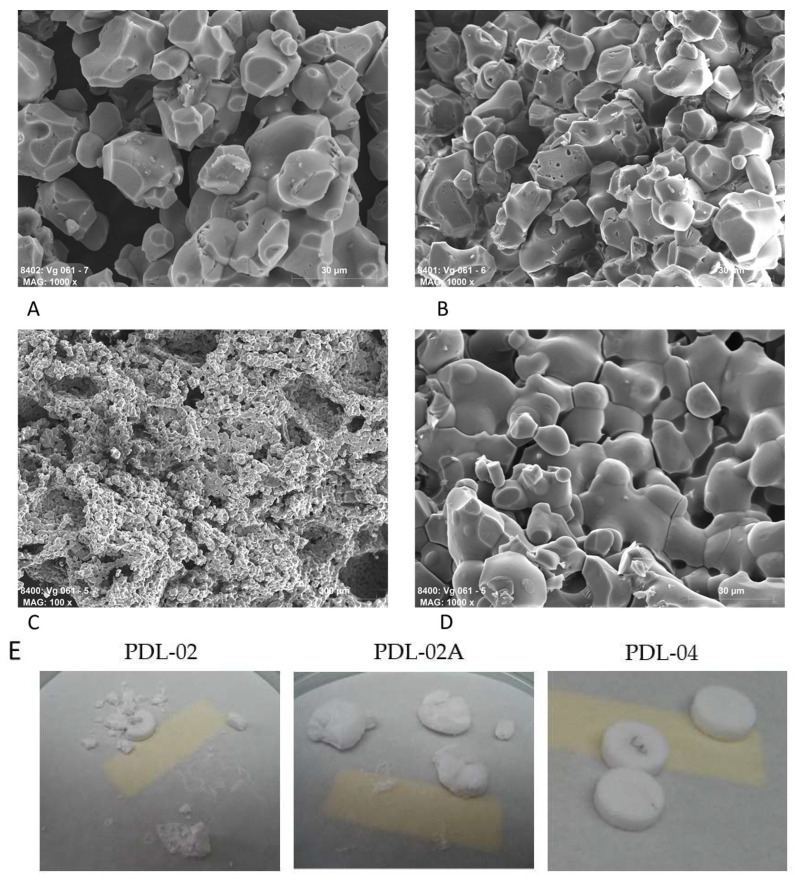
SEM photos of the polymer-coated β-TCP scaffolds after 180 days of degradation in deionized water. (**A**) PDL-02 (magnification 1000×); (**B**) PDL-02A (1000×); (**C**,**D**) PDL-04 (100× and 1000×); (**E**) Overview of three pieces of PDL-02, PDL-02A and PDL-04 β-TCP scaffolds after 180 days of degradation.

**Figure 6 jfb-14-00427-f006:**
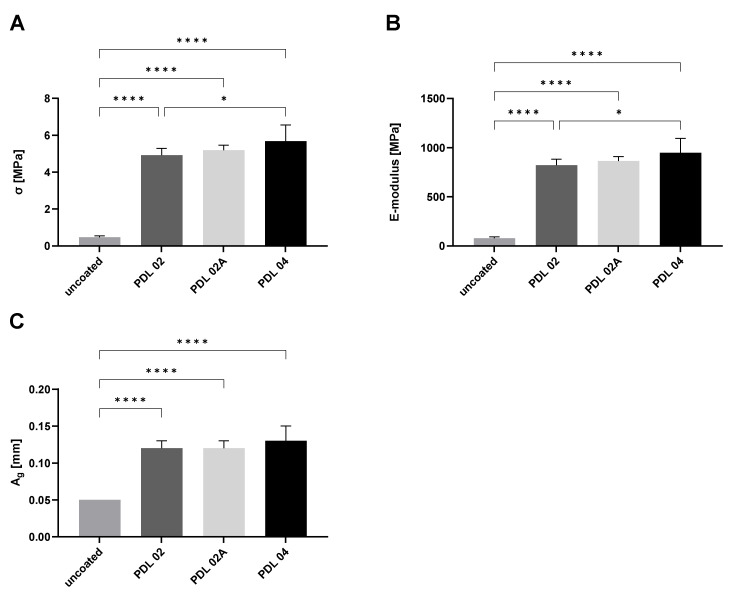
Mechanical properties of β-TCP variants used in this study. (**A**) Mean values and standard deviation of 3-point flexural strength of PLA-coated (PDL-02, -02A, -04) and uncoated β-TCP scaffolds; (**B**) mean values and standard deviation of the modulus of elasticity of PLA-coated (PDL-02, -02A, -04) and uncoated β-TCP scaffolds; (**C**) mean values and standard deviation of the bending elongation at break of PLA-coated (PDL-02, -02A, -04) and uncoated β-TCP scaffolds. The samples (*n* = 6 for uncoated and *n* = 8 coated specimens per group) were compared using one-way ANOVA and Tuckey’s multiple comparison test (* *p* ≤ 0.05, **** *p* ≤ 0.0001).

**Figure 7 jfb-14-00427-f007:**
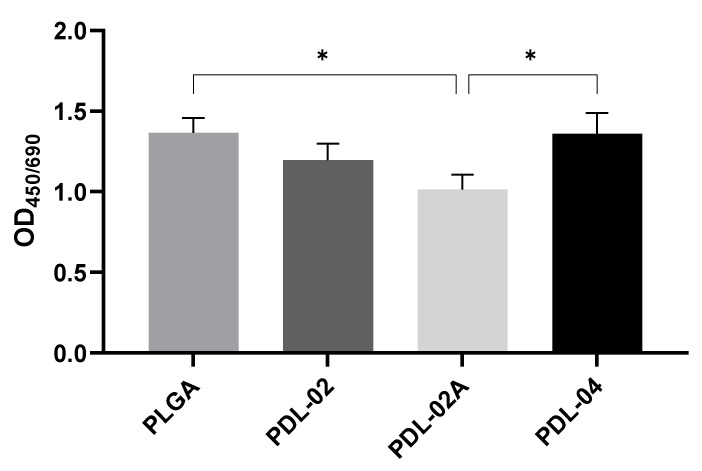
Metabolic activities of JPCs grown for 8 days on β-TCP scaffolds coated with four different PLA polymers (PLGA (internal control), PDL-02, PDL-02A, PDL-04). OD_450/690_ values are displayed as means ± SD (*n* = 3 donors) and were compared using one-way ANOVA with Tuckey’s multiple comparison test (* *p* < 0.05).

**Figure 8 jfb-14-00427-f008:**
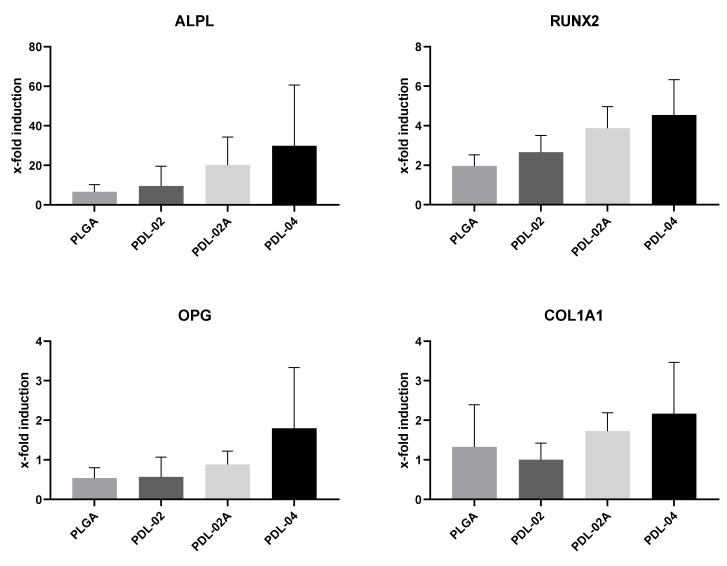
Gene expression analyses of JPCs grown for 8 days on β-TCP scaffolds coated with four different PLA polymers (PLGA, PDL-02, PDL-02A, PDL-04) in osteogenic medium. ALPL, RUNX2, OPG and COL1A1 mRNA copy numbers were detected and normalized to those of the housekeeping gene GAPDH. x-fold induction was calculated in relation to the corresponding untreated control samples. Values are displayed as means ± SD (*n* = 3 donors) and were compared using one-way ANOVA with Tuckey’s multiple comparison test.

**Figure 9 jfb-14-00427-f009:**
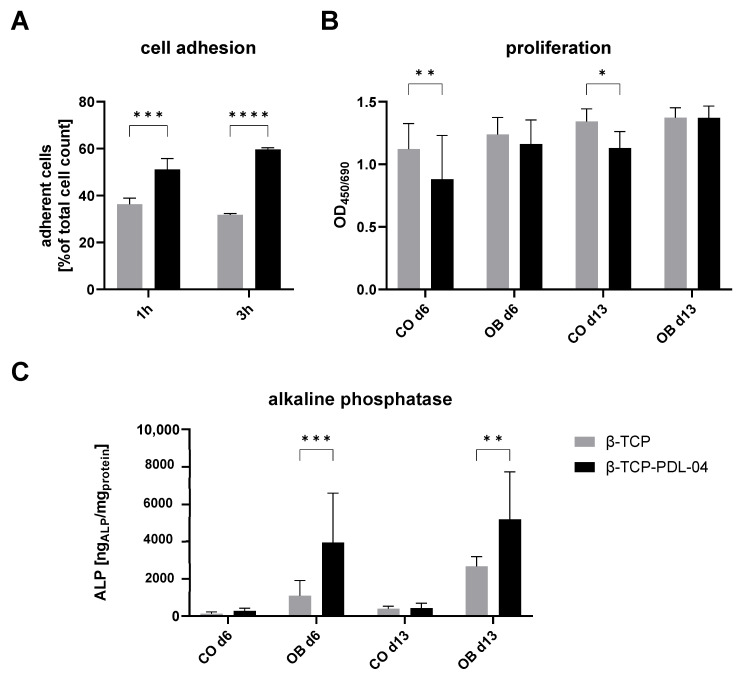
Adhesion, proliferation and alkaline phosphatase expression of JPCs growing on PDL-04-coated (black) and uncoated (grey) β-TCP scaffolds. (**A**) Cell adhesion: Percentages of adherent cells related to the total cell number after one and three hours; (**B**) cell proliferation activity: JPCs were cultured with control (CO) and osteogenic medium (OB) for 6 and 13 days; (**C**) alkaline phosphatase activity: JPCs were cultured with control (CO) and osteogenic medium (OB) for 6 and 13 days. Alkaline phosphatase activity was normalized to whole protein concentration in the same samples. Values are displayed as means ± SD (*n* ≥ 3 samples) and compared using two-way ANOVA with Šídák’s multiple comparisons test (* *p* < 0.05, ** *p* < 0.01, *** *p* < 0.001, **** *p* < 0.0001).

**Figure 10 jfb-14-00427-f010:**
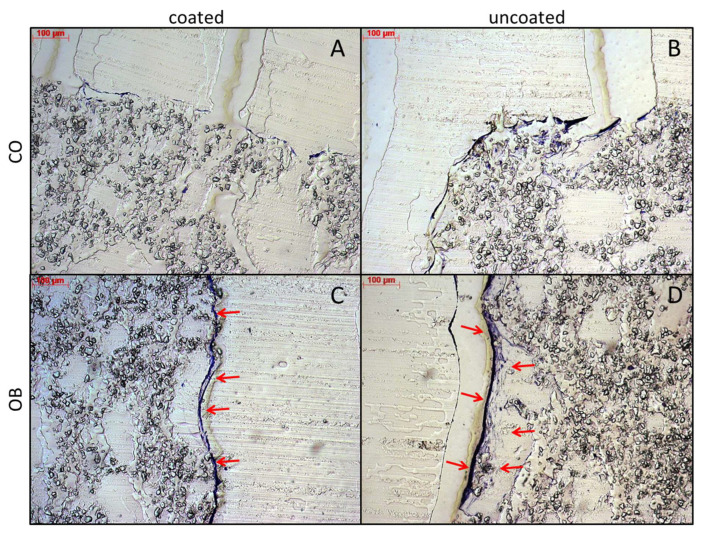
Toluidine blue staining of PDL-04 coated (**A**,**C**) and uncoated (**B**,**D**) β-TCP scaffolds. Cell-seeded scaffolds were treated with control (CO) and osteogenic (OB) media for 17 days. 5 µm sections of PMMA-embedded scaffolds were stained with toluidine blue and brightfield images were taken at 10-fold magnification (scale bar = 100 µm). Cell layers are highlighted by red arrows.

**Figure 11 jfb-14-00427-f011:**
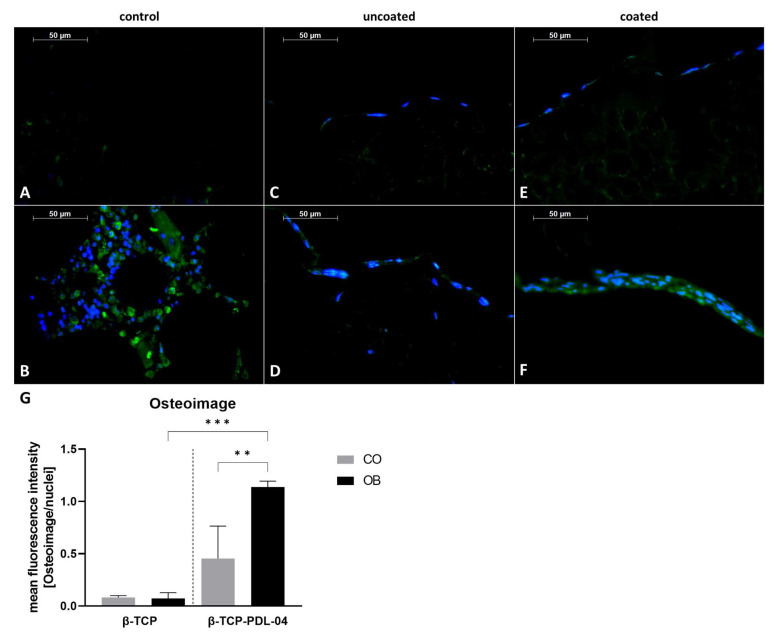
Detection of hydroxyapatite formation by OsteoImage staining (green) and nuclear counter staining by Hoechst 33342 (blue). Stained 5 µm sections of (**A**) PDL-04 coated control scaffold without cells (negative control); (**B**) human fibula (positive control); (**C**) uncoated JPC-seeded β-TCP scaffold cultured in control medium; (**D**) uncoated JPC-seeded β-TCP scaffold cultured in osteogenic medium; (**E**) PDL-04 coated JPC-seeded β-TCP scaffold cultured in control medium; (**F**) PDL-04 coated JPC-seeded β-TCP scaffold cultured in osteogenic medium for 17 days (fluorescence images were taken with a 10-fold magnification, scale bar = 50 µm); (**G**) quantification of fluorescence intensity by image analysis. Fluorescence intensity of OsteoImage staining was quantified using ImageJ software and normalized to the intensity of Hoechst33342 nuclear staining. Mean values were compared using one-way ANOVA and Tuckey’s multiple comparison test (** *p* < 0.01, *** *p* < 0.001).

**Figure 12 jfb-14-00427-f012:**
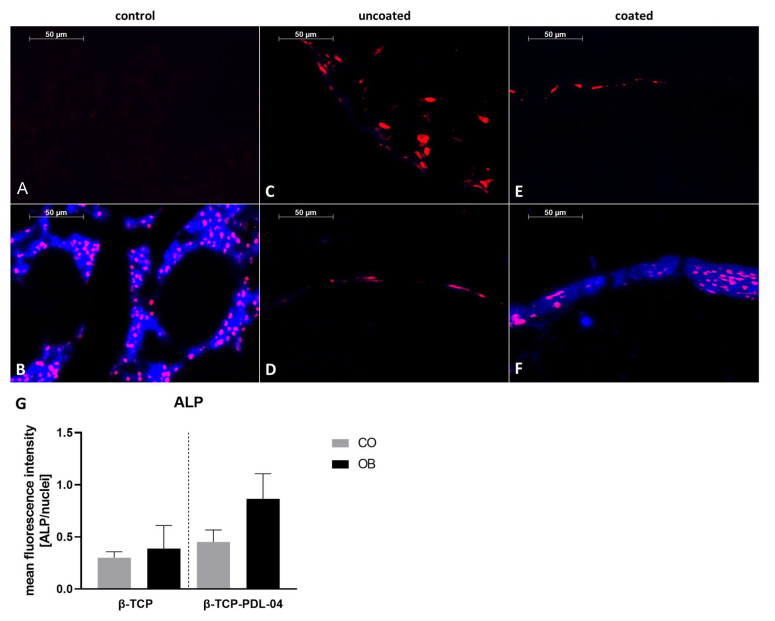
Detection of alkaline phosphatase expression with anti-ALP and DyLight405 antibodies (blue) and SYTOX Orange nuclear counter staining (red). Stained 5 µm sections of (**A**) PDL-04 coated control scaffold without cells (negative control); (**B**) human fibula (positive control); (**C**) uncoated JPC-seeded β-TCP scaffold cultured in control medium; (**D**) uncoated JPC-seeded β-TCP scaffold cultured in osteogenic medium; (**E**) PDL-04 coated JPC-seeded β-TCP scaffold cultured in control medium; (**F**) PDL-04 coated JPC-seeded β-TCP scaffold cultured in osteogenic medium for 17 days (fluorescence images were taken with a 10-fold magnification, scale bar = 50 µm); (**G**) quantification of fluorescence intensity by image analysis. Fluorescence intensity of ALP immunostaining was quantified using ImageJ software and normalized to the intensity of SYTOX Orange nuclear staining. Mean values were compared using one-way ANOVA and Tuckey’s multiple comparison test.

**Figure 13 jfb-14-00427-f013:**
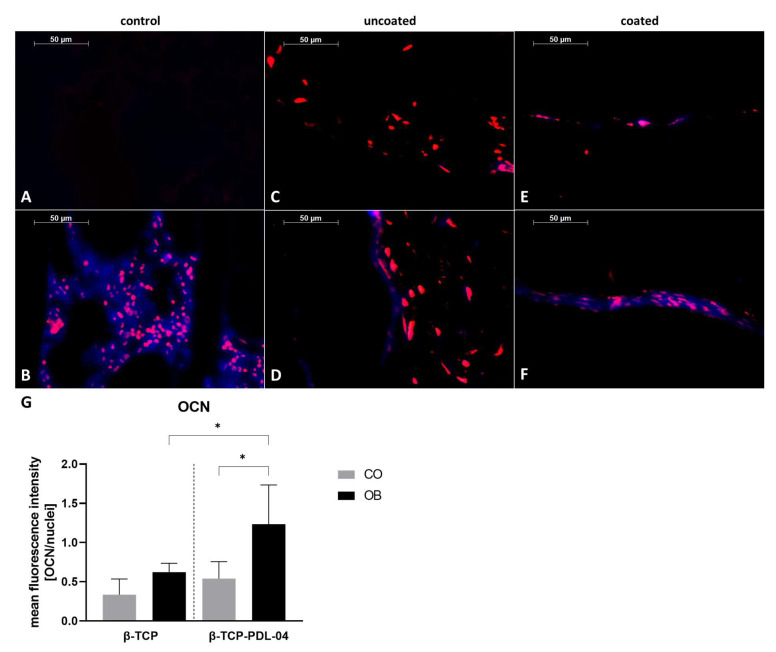
Detection of osteocalcin expression using anti-OCN and DyLight405 antibodies and nuclear counter staining by SYTOX Orange (red). Stained 5 µm sections of (**A**) PDL-04 coated control scaffold without cells (negative control); (**B**) human fibula (positive control); (**C**) uncoated JPC-seeded β-TCP scaffold cultured in control medium; (**D**) uncoated JPC-seeded β-TCP scaffold cultured in osteogenic medium; (**E**) PDL-04 coated JPC-seeded β-TCP scaffold cultured in control medium; (**F**) PDL-04 coated JPC-seeded β-TCP scaffold cultured in osteogenic medium for 17 days (fluorescence images were taken with a 10-fold magnification, scale bar = 50 µm); (**G**) quantification of fluorescence intensity by image analysis. Fluorescence intensity of OCN immunostaining was quantified using ImageJ software and normalized to the intensity of SYTOX Orange nuclear staining. Mean values were compared using one-way ANOVA and Tuckey’s multiple comparison test (* *p* < 0.05).

**Figure 14 jfb-14-00427-f014:**
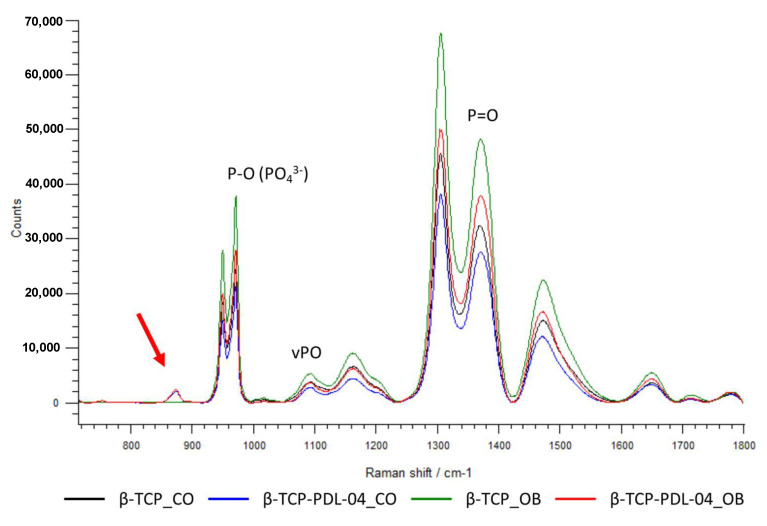
Representative averaged Raman spectra. Comparison of cell-free uncoated and PDL-04 coated β-TCP scaffolds incubated for 17 days with osteogenic (OB) and control (CO) medium. The averaged spectra were baseline corrected. A characteristic peak for the PLA coating at 874 cm^−1^ is highlighted with the red arrow.

**Figure 15 jfb-14-00427-f015:**
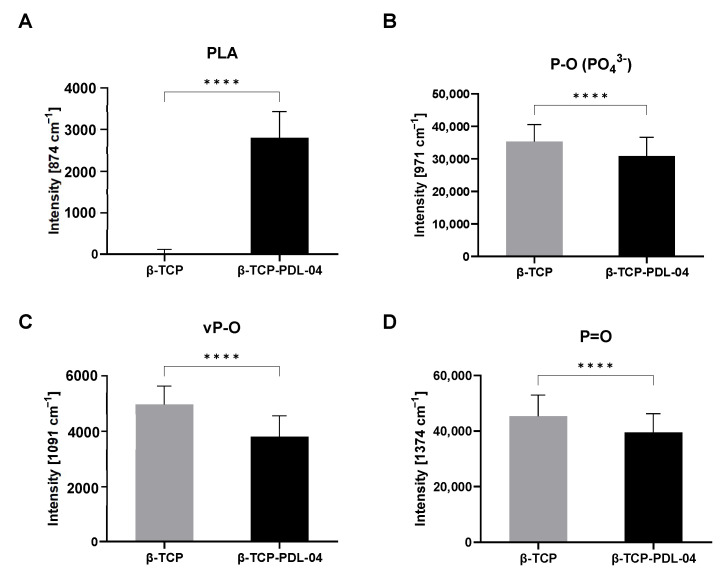
Raman analysis of cell-free scaffolds in osteogenic medium (OB). Intensities of specific Raman shift peaks from mean Raman spectra of cell-free uncoated (β-TCP) and PLA-coated (β-TCP-PDL-04) β-TCP scaffolds: (**A**) PLA peak (874 cm^−1^), (**B**) P-O (PO_4_^3−^) peak (971 cm^−1^), (**C**) νP-O peak (1091 cm^−1^), (**D**) P=O peak (1374 cm^−1^). Intensities are displayed as means + SD (three samples, 144 measurements per sample), and are compared using unpaired *t*-tests (**** *p* < 0.0001).

**Figure 16 jfb-14-00427-f016:**
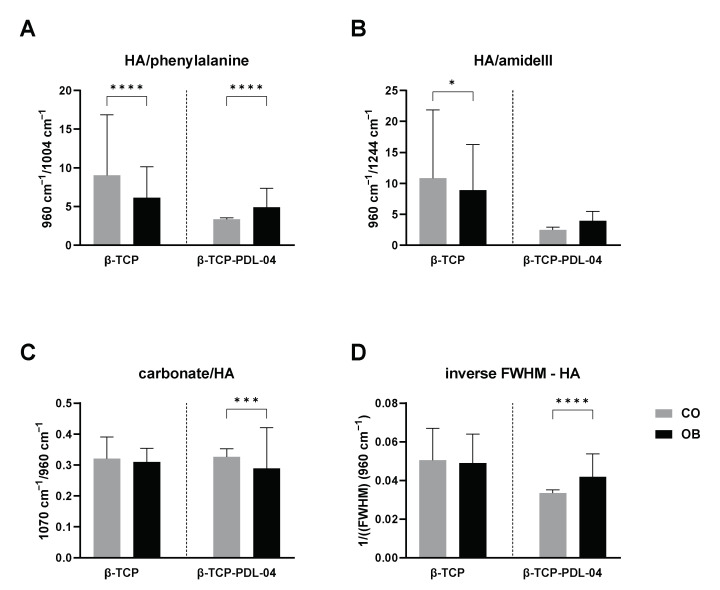
Ratios calculated from mean Raman spectra of cell-seeded uncoated (β-TCP) and PLA-coated (β-TCP-PDL-04) β-TCP scaffolds. Cell-seeded scaffolds were treated with control (CO) and osteogenic (OB) medium for 17 days. (**A**) HA to phenylalanine ratio; (**B**) HA to amid III ratio; (**C**) carbonate to HA ratio; (**D**) inverse full-width half maximum (1/FWHM) of the HA peak was calculated as a measure for HA crystal size. Values are displayed as means + SD (three donors, 90–148 measurements per donor), and are compared using two-way ANOVA with Tuckey’s multiple comparison correction (* *p* < 0.05, *** *p* < 0.001, **** *p* < 0.0001).

**Table 1 jfb-14-00427-t001:** Characteristics of the examined polymers.

Type PURASORB	End Group	Inherent Viscosity in CHCl_3_ [dL/g]
PDL-02	Ester terminated	0.2
PDL-02a	Acid terminated	0.2
PDL-04	Ester terminated	0.4

## Data Availability

The data that support the findings of this study are available from the corresponding author upon reasonable request.

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
