# Peer review of "Mechanical and Functional Improvement of β-TCP Scaffolds for Use in Bone Tissue Engineering"

_jfb, 2023, doi:10.3390/jfb14080427_

Round 1
Reviewer 1 Report
The article is well described, the introduction gathers appropriate elements for understanding the proposed theme, and the methodology is adequate. I recommend accepting the article with some observations.
Materials and Methods
11- I believe that the text between lines 83 and 97 is from the MDPI group template. Please, authors, confirm and exclude it.
22- Remove 'Error! Reference source not found..' from line 144.
33- Standardize the values mentioned in the text according to the International System of Units.
44- Line 287 - 3.1. Polymer coating results 287 - The authors need to provide images that prove the presence of micropores.
55- Line 296 - '(Error! Reference source not found.).'
66- Line 296 - What would be the appropriate degradation time for the authors' proposal?
77- Why didn't the authors include images of SEM without gamma radiation sterilization? Wouldn't it be interesting to observe the differences?
8.8- The authors need to add images of the membranes and discuss their visual characteristics.
99- Check for errors throughout the PDF references.
110. What is the indicated or 'ideal' value of elastic modulus for the scaffolds?
Author Response
11- I believe that the text between lines 83 and 97 is from the MDPI group template. Please, authors, confirm and exclude it.
We apologize this mistake. We removed the respective paragraph.
22- Remove 'Error! Reference source not found..' from line 144.
In the revised version of the manuscript this error should not occur any longer.
33- Standardize the values mentioned in the text according to the International System of Units.
The revised version of the manuscript contains values according to the International System of Units.
44- Line 287 - 3.1. Polymer coating results 287 - The authors need to provide images that prove the presence of micropores.
Unfortunately, we performed SEM images only after degradation. But the presence of micropores was proven by the determination of the water absorption capacity. These results are now given in the new table 3. PDL-04 coated scaffolds showed a significant lower water resorption capacity compared to PDL-02 coated scaffolds. Therefore, we assume that the micropores were of a smaller size on PDL-04 coated scaffolds. We mentioned this aspect now also in the Results and Discussion part.
55- Line 296 - '(Error! Reference source not found.).'
This error should not occur anymore in the revised manuscript.
66- Line 296 - What would be the appropriate degradation time for the authors' proposal?
Appropriate degradation time should be 6-12 months simultaneously to new bone formation with suitable mechanical characteristics through the complete degradation time.
77- Why didn't the authors include images of SEM without gamma radiation sterilization? Wouldn't it be interesting to observe the differences?
We made the observation that gamma radiation affects the polymer substrate only at the molecular but not microscopic level. But since the scaffold surface after gamma sterilization is the relevant surface with which cells interact, we decided to include only these SEM images.
8.8- The authors need to add images of the membranes and discuss their visual characteristics.
Through the optimization of the coating protocol, polymer coating is nearly not visible and penetrates the scaffolds also in deeper layers. Therefore, visual characteristics are hardly to distinguish.
But we added in the revised manuscript (Table 2, Results section) the measurements that we performed to calculate the coating efficiency for the PDL-04 coated β-TCP scaffolds. We included only the data from PDL-04 scaffolds, since they are the only relevant for the performed cell experiments.
Beside this, we added part E of Figure 5, in order to give an overview of degraded coated composite samples after 180 days.
99- Check for errors throughout the PDF references.
This error should not occur anymore in the revised manuscript.
- What is the indicated or 'ideal' value of elastic modulus for the scaffolds?
The “ideal” value would be the elastic modulus of natural bone that can never be reached with artificial inverse composite materials of polymer-coated ceramics. The coating can reach improved elasticity modules by enhancing the flexibility of the scaffold by simultaneously reducing the brittleness.
Reviewer 2 Report
Reviewer’s comments:
In the following research article ‘Mechanical and functional improvement of β-TCP scaffolds for use in bone tissue engineering,’ the authors successfully surface-modified β-TCP scaffolds and explored their activities in bone tissue regeneration. The article has comprehensively included a wide variety of characterization techniques, degradation kinetics, applications. Here are some comments for the authors:
1. The first three paragraphs of materials and methods make no sense. Those are copied from the journal policy. Please be careful while you are submitting a manuscript.
2. The manuscript is full of “Error! Reference source not found” statements (lines 144, 178, 296, 307, 319, 322, 328, 332, 339, 352, 357, 360, 402, 418, 433, 434, 435, 436, 439, 440, 449, 452, 454, 456, 458, 459, 461, 482, 491, 507, 510, 514, 515, 522).
3. The references pointing to the figures are missing in many places (e.g., line 307). In some places, those are partially present (lines 319, 339, and many more) and not able to properly indicate the figure. It is written A-C (line 339) but the figure number is missing. The authors should check the manuscript thoroughly before re-submission. A thorough recheck for this point is really necessary.
4. Phosphate groups are PO43-. The authors have written it in the other way around. Please correct it wherever necessary.
5. After manufacturing, the surface-modified β-TCP scaffolds did not go through any spectroscopic characterization. We strongly suggest solid-state FT-IR characterization for confirming the presence of surface-modifying molecules on the scaffolds. Data collection after proper baseline correction with uncoated scaffolds should be able to provide the relevant signatures confirming the surface modification.
6. Water contact angle is an indication of surface hydrophilicity which in general leads to higher cellular adhesion. The contact angle values for each modified and control scaffolds should be reported along with images.
7. It is necessary to know, after exposure to water how much water is retained on the polymer-coated surfaces. A quantitative estimation of that can be indicated by a plot of the weight of water absorbed per unit weight of the scaffold.
8. Lines 149-154 talk about weighing the surface-modified β-TCP scaffolds. But the weight corresponding to each scaffold is not included in the manuscript. We suggest a QCM (quartz crystal microbalance) analysis to find out the exact weight of the surface-modifying polymers.
9. Figure 1 does not have any mention in the text of the manuscript.
10. The formula used for the 3-point bending strength has the letter ‘I’. However, in line 183 the authors explain the number ‘1’. Please correct if it is unintentional.
11. Figure 4C has a different magnification (50 μm) in comparison to 4A, B, and D (30 μm).
12. Figure 5C has a different magnification (300 μm) in comparison to 4A, B, and D (30 μm).
13. In Figure 7, why is the uncoated control sample β-TCP is not included?
14. In section 3.7 line 404, there should be a space between ‘both and 1h’. A comma in between is misleading.
15. ‘Alkaline phosphatase’ is sometimes written as ALP and sometimes as ALPL. The authors should be consistent.
16. How is the percent adhesion (figure 9C) calculated?
17. What is the positive control in the adhesion experiment?
18. In Figure 10, please name each panel with A, B, C, and D. Include these in legends and explicitly discuss.
19. With the help of Figure 10, submit an average of random cell count data indicating the overall cellular population.
20. For the detection of mineralization EDAX and ICP-MS can be performed on the scaffold surface. Elemental mapping from SEM can indicate mineralization.
Re-reading and some grammatical corrections are necessary.
Author Response
- The first three paragraphs of materials and methods make no sense. Those are copied from the journal policy. Please be careful while you are submitting a manuscript.
We apologize this stupid mistake. We removed these paragraphs and now the section of materials and methods begins with “Cell isolation and culture of JPCs”.
- The manuscript is full of “Error! Reference source not found” statements (lines 144, 178, 296, 307, 319, 322, 328, 332, 339, 352, 357, 360, 402, 418, 433, 434, 435, 436, 439, 440, 449, 452, 454, 456, 458, 459, 461, 482, 491, 507, 510, 514, 515, 522).
The version jfb-2486570 that we downloaded from the journal’s homepage does not contain any “Error! Reference source not found” statement. We used this version for revision.
- The references pointing to the figures are missing in many places (e.g., line 307). In some places, those are partially present (lines 319, 339, and many more) and not able to properly indicate the figure. It is written A-C (line 339) but the figure number is missing. The authors should check the manuscript thoroughly before re-submission. A thorough recheck for this point is really necessary.
In the revised version, figure number is given wherever necessary. They are all marked in red.
- Phosphate groups are PO43-. The authors have written it in the other way around. Please correct it wherever necessary.
We corrected the designation for phosphate groups.
- After manufacturing, the surface-modified β-TCP scaffolds did not go through any spectroscopic characterization. We strongly suggest solid-state FT-IR characterization for confirming the presence of surface-modifying molecules on the scaffolds. Data collection after proper baseline correction with uncoated scaffolds should be able to provide the relevant signatures confirming the surface modification.
Thanks for this suggestion. Unfortunately, we are not able to perform solid-state FT-IR. But in our opinion, Raman spectroscopic analyses are enough to confirm PLA coating at the specific peak 874 cm-1 as illustrated in Figure 14 and as reported also by several other studies (references 23, 24).
- Water contact angle is an indication of surface hydrophilicity which in general leads to higher cellular adhesion. The contact angle values for each modified and control scaffolds should be reported along with images.
Thanks for this suggestion. We are not able to perform contact angle measurements with a new lot number. This needs a long time for establishing.
But in a previous study from our lab (reference 10), we performed water contact angle analyses with uncoated and PLGA-coated β-TCP scaffolds. Both scaffold types showed a hydrophobic surface with no significant difference. Despite of the fact that hydrophobic surfaces are supposed to be not attractive for cells, we cannot confirm this point neither in the present work nor in previous studies. We considered this aspect which is now mentioned in the discussion part of the revised manuscript.
- It is necessary to know, after exposure to water how much water is retained on the polymer-coated surfaces. A quantitative estimation of that can be indicated by a plot of the weight of water absorbed per unit weight of the scaffold.
Thanks for this suggestion. We added a new table (Table 3) with obtained data of the analysis of the water absorption capacity. However, we were not able to determine WAC of PDL-02a coated scaffolds.
- Lines 149-154 talk about weighing the surface-modified β-TCP scaffolds. But the weight corresponding to each scaffold is not included in the manuscript. We suggest a QCM (quartz crystal microbalance) analysis to find out the exact weight of the surface-modifying polymers.
Thanks for this suggestion. In our opinion, the analyses by QCM will not exactly reflect the real condition and topography of herein used scaffolds to obtain reliable data of polymer uptake.
We added a new table with the data of the evaluation of the PDL-04 polymer uptake/coating efficiency in the part of scaffold manufacturing of the Materials/Methods section (Table 2).
- Figure 1 does not have any mention in the text of the manuscript.
Now, figure 1 is mentioned in the section materials/methods.
- The formula used for the 3-point bending strength has the letter ‘I’. However, in line 183 the authors explain the number ‘1’. Please correct if it is unintentional.
Thanks for the note, corrected
- Figure 4C has a different magnification (50 μm) in comparison to 4A, B, and D (30 μm).
This is correct since A illustrates PDL-02, B illustrates PDL-02A and C and D illustrates PDL-04 in a smaller and larger magnification.
- Figure 5C has a different magnification (300 μm) in comparison to 4A, B, and D (30 μm).
This is correct since A illustrates PDL-02, B illustrates PDL-02A and C and D illustrates PDL-04 coated β-TCP scaffolds in a smaller and larger magnification.
- In Figure 7, why is the uncoated control sample β-TCP is not included?
For the analysis of JPC proliferative activities, we wanted to find the best suitable coating for the cells. Therefore, we compared only the coated scaffold groups which other.
- In section 3.7 line 404, there should be a space between ‘both and 1h’. A comma in between is misleading.
Thanks for the note, corrected
- ‘Alkaline phosphatase’ is sometimes written as ALP and sometimes as ALPL. The authors should be consistent.
We use the abbreviation ALPL only for gene expression because this is the reported designation in the human gene database (Gene cards). For protein expression, we use the abbreviation ALP.
- How is the percent adhesion (figure 9C) calculated?
We added a paragraph to the Materials/Methods part explaining the cell adhesion experiments.
- What is the positive control in the adhesion experiment?
Since we were interested in the comparison of coated/uncoated scaffold groups, we did not consider a positive control. The percentage of adherent cells was calculated relative to total cell number counts.
- In Figure 10, please name each panel with A, B, C, and D. Include these in legends and explicitly discuss.
Thanks for the note, corrected
- With the help of Figure 10, submit an average of random cell count data indicating the overall cellular population.
Unfortunately, we are not able to generate cell count data from the toluidine blue images. Stained cells formed a continuous cell layer and therefore single cell nuclei cannot be visualized.
- For the detection of mineralization EDAX and ICP-MS can be performed on the scaffold surface. Elemental mapping from SEM can indicate mineralization.
Thanks for this suggestion. Unfortunately, we are not able to perform EDAX and ICP-MS. But we think that fluorescent staining (OsteoImage), immunofluorescent images (ALP and osteocalcin) and Raman data are convincing enough to prove cell mineralization. Additionally, we now added the quantifications of these stainings (part G of Fig. 11, 12, 13).
- Comments on the Quality of English Language: Re-reading and some grammatical corrections are necessary.
Thank you, we proof read the whole manuscript.
Reviewer 3 Report
This paper by Felix Umrath et al. reported a study on the improvement of β-TCP scaffolds in bone tissue engineering. Due to the following reasons, I recommend acceptance of this manuscript for publication after the revision.
1. Please explain and revise the content "Error! Reference source not found" of line 178, line 296, line 319, line 322 et al in the full article.
2. In the experiment 3.5 part, please add Live/dead staining or EDU to supplement cell proliferation.
3. In the experiment 3.7 section, please add the digital images of the whole holes of ALP.
4. Please supplement Alizarin Red staining to enhance the convincingness of osteogenesis.
5. Please make a statistical analysis for Fig10, Fig11, Fig12, and Fig13.
Author Response
- Please explain and revise the content "Error! Reference source not found" of line 178, line 296, line 319, line 322 et al in the full article.
The version jfb-2486570 that we downloaded from the journal’s homepage does not contain any “Error! Reference source not found” statement. We used this version for revision.
- In the experiment 3.5 part, please add Live/dead staining or EDU to supplement cell proliferation.
Thanks for this suggestion. Unfortunately, we are not able to perform new experiments because these will include a new scaffold lot number. In our opinion, it is difficult to compare the other results from the manuscript from the first scaffold lot number with those (EDU or live/dead staining) from a new scaffold lot number.
- In the experiment 3.7 section, please add the digital images of the whole holes of ALP.
Unfortunately, we don’t understand what is required here.
- Please supplement Alizarin Red staining to enhance the convincingness of osteogenesis.
Thanks for this suggestion. The Alizarin red staining is not optimal to detect mineralization within β-TCP scaffolds because the dye binds strongly and unspecifically to the core material. Nevertheless, we made the experience that a comparison of uncoated cell-seeded β-TCP scaffolds under untreated and osteogenic condition is quantifiable.
However, for the comparison of uncoated with coated β-TCP scaffolds, Alizarin staining is very imprecise. Firstly, the dye binds to the core material and to the minerals formed by the cells. Secondly, coated scaffolds bind the dye to a much lower extend due to the coating which covers not only phosphate but also calcium sites of the core material. Due to these aspects at the end, we cannot obtain a reliable quantification.
Therefore, we used more reliable methods such as gene expression, Raman spectroscopy and immunofluorescent staining to detect successful osteogenic differentiation. Besides of this, we now added the quantification of the fluorescent/immunofluorescent staining illustrated in Fig. 11, 12, 13 which provide further evidence of a convincing osteogenesis.
- Please make a statistical analysis for Fig10, Fig11, Fig12, and Fig13.
Thanks for this recommendation. We added quantification data (part G of Fig. 11, 12, 13) for the fluorescent OsteoImage (mineralization) staining, ALP and osteocalcin immunofluorescent staining.
Concerning the Figure 10, we were not able to perform quantification, or better said quantification data are not reliable, based on the fact that no single cells/nuclei but only a continuous, stained cell layer can be visualized.
Reviewer 4 Report
The given study was conducted on β-tricalcium phosphate scaffolds coated with polymers (PLA) for bone tissue engineering applications. The study is well designed and results are interesting.
However, I have few questions/suggestions to the authors;
1. Why authors have selected 3-point bend test for mechanical analysis. This should be clearly specified in the manuscript.
2. In abstract, authors have given comparison in terms of compressive strength. Is this bending strength? Or how they calculated the compressive strength through bending?
3. Materials and Method line 82 to 97: I am confused if this information is provided by the authors.
4. As the sample size is small (n =6), did authors perform normality test and if level of significance 0.05 good enough. I would suggest nonparametric test with p = 0.01 for comparing the mechanical results.
Author Response
- Why authors have selected 3-point bend test for mechanical analysis. This should be clearly specified in the manuscript.
In general, the flexural strength test tends to be more sensitive in assessing the performance of bone substitutes for certain dental applications. This is because they often experience bending forces during functional stresses like chewing and biting. Furthermore, we chose the 3-point bending test because it was our goal to investigate whether the coatings improved the elasticity. This rationale for the test choice is now stated in the Discussion section according to your advice.
- In abstract, authors have given comparison in terms of compressive strength. Is this bending strength? Or how they calculated the compressive strength through bending?
Thank you for pointing this out. Indeed, we have studied the flexural strength and not the compressive strength, as it is mistakenly mentioned in the abstract. The misspelling has been corrected in the manuscript.
- Materials and Method line 82 to 97: I am confused if this information is provided by the authors.
We apologize this mistake. We removed the respective paragraph.
- As the sample size is small (n =6), did authors perform normality test and if level of significance 0.05 good enough. I would suggest nonparametric test with p = 0.01 for comparing the mechanical results.
The samples were compared using one-way ANOVA and Tuckey’s multiple comparison test (* = p ≤ 0.05, **** = p ≤ 0.0001). This was corrected in the statistical analysis section. The sample size is common for such mechanical tests and sufficient for the statistical power of the test to detect significant differences between material variants, which is also indicated by the low standard deviation. We tested 6 uncoated samples and 8 coated specimens per group. We made the corrections in the Materials/Methods part.
Reviewer 5 Report
abstract needs to be revised, clear statement of the study's main aim based on the available literature has to be identified.
introduction: it is too wordy, and the first paragraph seems unnecessary.
materials and method: this section is no sense and must be changed.
discussion: this section is missing the citation and comparison of the available literature. what has been done in the literature to improve the mechanical properties of the Beta-TCP? what are the current pitfalls?
minor revision is required
Author Response
Comments and Suggestions for Authors
abstract needs to be revised, clear statement of the study's main aim based on the available literature has to be identified.
Thanks for this suggestion. The abstract was rewritten in order to clearly state the study’s main aim.
introduction: it is too wordy, and the first paragraph seems unnecessary.
The first paragraph was removed.
materials and method: this section is no sense and must be changed.
Thanks for this suggestion. We removed the unnecessary part of the Materials/Methods section and added the missing part concerning cell adherence in the revised manuscript.
discussion: this section is missing the citation and comparison of the available literature. what has been done in the literature to improve the mechanical properties of the Beta-TCP? what are the current pitfalls?
Thanks for this suggestion. We added a large paragraph including the citation of previous studies dealing with attempts to improve the mechanical properties of CaP materials explaining the pitfalls and challenges.
Comments on the Quality of English Language
minor revision is required
We revised the English language in the whole manuscript.
Round 2
Reviewer 2 Report
1. Line 122, 176, 315, 316, 328, 331, 334, 342, 344, 346, 356, 360, 373, 386, 391, 395, 437, 442, 450, 452, 459, 469, 471, 473, 477, 478, 493, 497, 499, 503, 505, 507, 509, 538, 547, 565, 568, 572, 574, 582, 584 – sentence appears to be incomplete with the error message ‘Fehler! Verweisquelle konnte nicht gefun-den werden’. Please clarify.
2. Line 122 – sentence is incomplete. It should be ‘Table 1’.
3. Line 161 – In SEM full form only S is capital.
4. Thorough English language checking is required. Especially, article checking is necessary.

Extensive editing is needed, especially the use of the article 'the'.
Author Response
- Line 122, 176, 315, 316, 328, 331, 334, 342, 344, 346, 356, 360, 373, 386, 391, 395, 437, 442, 450, 452, 459, 469, 471, 473, 477, 478, 493, 497, 499, 503, 505, 507, 509, 538, 547, 565, 568, 572, 574, 582, 584 – sentence appears to be incomplete with the error message ‘Fehler! Verweisquelle konnte nicht gefun-den werden’. Please clarify.
These errors exist only in the pdf file, after converting the word file into the pdf.
We hope that now the error does not occur anymore.
- Line 122 – sentence is incomplete. It should be ‘Table 1’.
This is the same error as mentioned in point 1. It should be deleted now.
- Line 161 – In SEM full form only S is capital.
We corrected and wrote the entire full form in small letters.
- Thorough English language checking is required. Especially, article checking is necessary.
We checked the English language in the whole article again.
Thanks so much for your time :-)
Reviewer 3 Report
The authors have addressed all my concerns. And the revised manuscript has been significantly improved.
Author Response
Thanks so much for your time and your constructive recommendations :-)
Reviewer 5 Report
the revision is satisfactory.
quality of English language is satisfactory
Author Response
Thanks so much for your time and your recommendations :-)